# Research on Dynamic Reserve and Energy Arbitrage of Energy Storage System

**Jia-Zhang Jhan [1], Tzu-Ching Tai [2], Pei-Ying Chen [1] and Cheng-Chien Kuo [1,*]**

1. Department of Electrical Engineering, National Taiwan University of Science and Technology, Taipei 10607, Taiwan
2. Graduate Institute of Energy and Sustainability Technology, National Taiwan University of Science and Technology, Taipei 10607, Taiwan
* Correspondence: cckuo@ntust.edu.tw

**Abstract:** Replacing the traditional rotating generators with renewable energy will reduce the grid's inertia and with it the minimum frequency when N-1 contingency occurs triggering an Under-Frequency Load Shedding (UFLS). This study proposes a method for the energy storage system (ESS) to simultaneously provide energy arbitrage, reserve capacity, and assist N-1 contingency, by modifying the restriction formula of economic dispatch (ED) and limiting the SOC range of the ESS. Let the ESS join the Spinning Reserve. Through the PSS®E iterative ESS charging power required at moments when the frequency of contingency is too low in the ED. Let the ESS act as a N-1 contingency extra frequency reserve. This would prevent UFLS and still maintain the demand. The proposed method is applicable to different types of ESS. The method allows energy storages, originally designed for energy arbitrage, to participate in frequency support and spinning reserve.

**Keywords:** energy storage system; renewable energy; economic dispatch; security constraint; PSS®E

## 1. Introduction

When the power generation and the power consumption are not equal, the frequency will deviate from the nominal value [1]. In the power system, if one of the generators fails, it will cause an energy mismatch and the frequency will begin to drop. The system needs to provide for the missing power generation immediately. There are frequency control responses that can be applied during generator contingency. First is the inertia response which comes from the rotating synchronous generator connected to the grid. When the frequency drops, the rotational inertia will be converted from kinetic energy into electrical energy and input into the grid instantly, reducing RoCoF (Rate of Change of Frequency). The second response is the droop control of some generators wherein the governor is controlled automatically to respond when the frequency starts to deviate [2]. This control allows more steam to enter the turbine to generate electricity, matching the grid energy for a few seconds, preventing frequency reduction, and using renewable energy instead of traditional rotation.

In the case of system energy imbalance, the rotational kinetic energy stored in the rotor of the traditional synchronous generator is used to provide inertial support for the power grid, keeping the minimum frequency of accidents at a certain level. However, most inverter-based sources (IBRs) such as wind or solar cannot provide inertia [3]. Replacing traditional generators with IBRs will reduce the inertia of the system, causing a larger RoCoF and a lower minimum frequency when the generator trips [4].

In the past, the power system usually used under-frequency load shedding (UFLS) to balance the insufficient power generation when the generator contingency caused the reduction of supply [4–7]. Some literatures [8–13] used RoCoF and the lowest frequency point to estimate the energy storage system (ESS) capacity and location for the frequency regulation required by the system.

In [10,14], the virtual inertia and the primary frequency response (PFR) provided by the ESS are taken into account to estimate the required ESS power and capacity. These two studies consider ESSs that can quickly respond to frequencies, but according to [15] not all ESSs have the ability to adjust the frequency.

In [16–18], ESS was added to the economic dispatch (ED) to deal with a high penetration of renewable energy. These two papers used ESS for peak shaving but did not consider frequency stability.

In [19], an ED considering automatic generation control (AGC) is proposed when the proportion of renewable energy is high, but peak shaving ESS is not considered. In [20], it is proposed to consider both PFR and secondary frequency response (SFR) in the unit commitment (UC) and consider the cost rate of ancillary services in different regions to propose a new market settlement strategy to compensate for the regional marginal cost of providing frequency reserves.

At present, most of the ED studies that consider security constraints do not include ESS. Further, most of the EDs that include ESS do not consider frequency security constraints. Therefore, this study proposes a method to incorporate both security constraints and ESS into the ED to find a safe and economical schedule.

In this study, the ESS was first added to the spinning reserve in the ED, which could reduce the generator's online time and thus reduce the cost. The base models are described in [21] with the addition of the ESS. With the inclusion of the ESS, the ESS will reduce the cost in the ED through its charging and discharging, and will automatically perform energy arbitrage, thereby providing two functions at the same time. However, due to the high penetration of renewable energy, the occurrence of N-1 contingency during certain periods will result in very low frequency. It is, therefore, important to calculate how much the ESS needs to be charged during these periods.

When an N-1 contingency occurs, the charging of the ESS can be cut off immediately to compensate for insufficient power generation and not trigger the UFLS during the low frequency. The addition of security constraints in the ED would find a safe ED to schedule.

PSS®E is a software from Siemens widely used in the analysis of power systems [22–24]. The N-1 contingency minimum frequency was also calculated using PSS®E in papers [22,23] and was used in this study. PSS®E mainly uses the dynamic reduction method to calculate contingency minimum frequency [25,26].

The content of the paper is as follows: Section 2 describes the proposed strategy; Section 3 describes the constraints of ED; Section 4 describes the simulation scenarios; Section 5 shows the simulation results; Section 6 is the discussions; and Section 7 is the conclusion.

## 2. Proposed Preventive Control Strategy

The flowchart shown in Figure 1, shows the flow of the simulations for the proposed preventive control strategy. The hourly power generation and hourly forecasted data of renewable energy will be first read to calculate the net load. Then, using the MILP, an ED for the new generator and ESS will be determined for the next 24 h. The scheduled ED will be used to determine the minimum frequency ($F_{nadir}$) that will be calculated at each hour using the PSS®E software. If the minimum frequency is lower than the set value ($F_{min}$), then the charging power of the ESS will be calculated by adding 0.1 MW in that hour to meet the $F_{min}$ requirement. This charge will be added to the ED constraint and rescheduled until the minimum frequency per hour is higher than the set value. If the maximum charge of the ESS is reached, one generator will be added to the schedule and the ED will again be computed. An additional charge will again be included in the ESS schedule to make sure that the new minimum frequency is greater than the set value. In other words, the minimum frequency of N-1 contingency should always be higher than the set value every hour to make sure that the charge from the ESS can support a sudden drop in frequency.

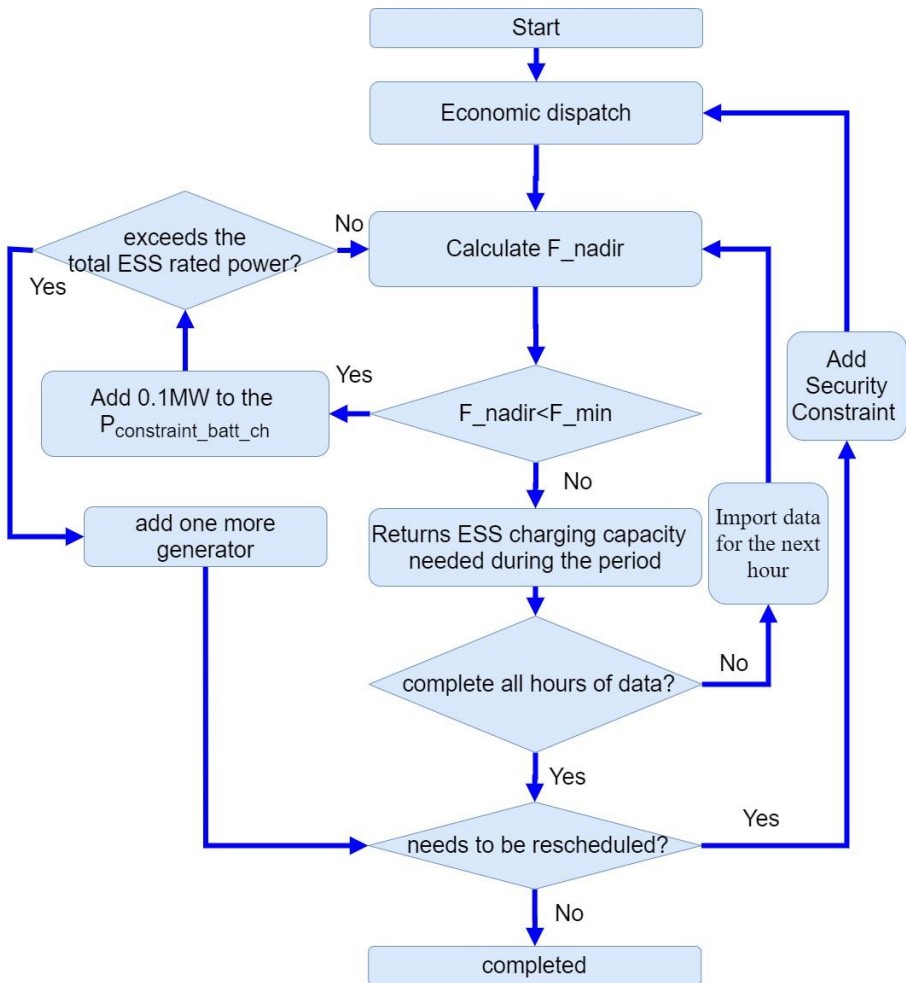

**Figure 1.** Flow chart.

## 3. Constraints of Economic Dispatch

### 3.1. Objective Function

The objective function is to minimize the operating cost, as shown in Equation (1). $C_L(t)$ represents the total fuel cost of all diesel generator sets at time $t$, $C_{st}(t)$ is the total startup cost of all diesel generator sets at time $t$, and $C_{batt}(t)$ represents the ESS cost. The power generation cost of PV in this study is set to 0. $F$ represents the total economic cost.

$$Min\ F = \sum_{t=1}^{24}[C_L(t) + C_{st}(t) + C_{batt}(t)] \tag{1}$$

### 3.2. Diesel Generator

Equation (2) indicates the total fuel cost of a diesel generator in quadratic form, where $FC_n$ represents the fuel cost of the nth diesel generator. The $a_n$, $b_n$ and $c_n$ represents the quadratic fuel cost constants of the $n$th diesel generator. $P_n(t)$ is the power generated by the $n$th diesel generator at time $t$. Figure 2 shows a typical fuel cost in quadratic form. However, because MILP is used, the quadratic curve needs to be linearized. In order to have a linear equation that is near the quadratic form, the curve is divided into segments and a line is drawn in each segment as the linear representation of the fuel cost curve for the $i$th segment.

$$FC_n(P_n(t)) = a_n + b_n P_n(t) + c_n P_n(t)^2 \tag{2}$$

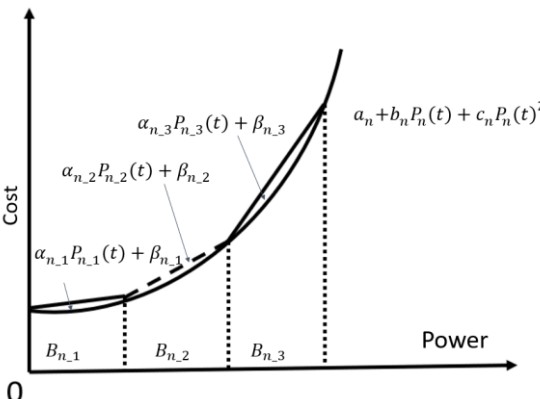

**Figure 2.** Typical generator cost curve with piecewise linearization.

Equations (3)–(5) are limits on the amount of electricity generated by the generator, where $P_{n\_i\_max}$ and $P_{n\_i\_min}$ are the maximum and minimum power generation in the $i$th line segment of the $n$ generator set, respectively. $B_{n\_i}(t)$ is a binary integer representing whether diesel generator $n$ is running in the $i$th linear interval at time $t$. $P_{n\_i}(t)$ is the amount of electricity generated in line segment $i$. The $i$ represents the number of the line segment in the quadratic curve. Inequality (5) ensures that only one line segment is selected for the $n$th generator at any given time $t$.

$$B_{n_i}(t)P_{n_{im}} \leq P_{n_i}(t) < B_{n\_i}(t)\,P_{n\_i\_max} \tag{3}$$

$$P_n(t) = \sum_{i=1}^{I} P_{n\_i}(t) \tag{4}$$

$$\sum_{i=1}^{I} B_{n_i}(t) \leq 1 \tag{5}$$

The total fuel cost of all generators set at time $t$ is expressed in Equation (6). $U_n(t)$ represents a binary integer variable of whether the $n$th diesel generator is turned on at time $t$. $N$ represents the total number of diesel generators. $\alpha_{n\_i}$ and $\beta_{n\_i}$ represent the slope and intercept, respectively, of the linear fuel cost when the $n$th diesel generator operates on line segment $i$ at time $t$.

$$C_L(t) = \sum_{n=1}^{N} \sum_{i=1}^{I} U_n(t)B_{n_i}(t) \left[\alpha_{n_i} P_{n_i}(t) + \beta_{n\_i}\right] \tag{6}$$

Equation (7) represents the total startup cost of $N$ diesel generators at time $t$. $ST_{price}$ represents the startup cost of generators.

$$C_{st}(t) = \sum_{n=1}^{N} ST_{price}(U_n(t) - U_n(t-1)) \tag{7}$$

*3.3. ESS*

Inequalities (8) and (9) limit the charge and discharge power of the ESS so that it does not exceed the limit of the maximum charge and discharge power. The $B_{batt\_dis}(t)$ and $B_{batt\_ch}(t)$ are binary integers representing the discharge or charge states of the ESS at time $t$ while. The $P_{batt\_dis\_max}$ and $P_{batt\_ch\_max}$ on the other hand represent the maximum discharge power and the maximum charging power of the battery ESS, respectively.

$$0 \leq P_{batt_{dis}}(t) \leq P_{batt\_dis\_max} B_{batt\_dis}(t) \tag{8}$$

$$-P_{batt\_ch\_max}B_{batt\_ch}(t) \leq P_{batt_{ch}}(t) \leq 0 \tag{9}$$

Inequality (10) is used to ensure that a single ESS cannot be charged and discharged at the same time.

$$B_{batt\_dis}(t) + B_{batt\_ch}(t) \leq 1 \tag{10}$$

Equation (11) describes the charge and discharge cost of the ESS. The purpose is to prevent the ESS from charging and discharging at an unnecessary time. Because if the battery ESS does not add the cost of charging and discharging. It may be charged and discharged in two time periods with the same electricity price. For example, if the average power generation cost of the first hour and the fifth hour is both 4. The battery ESS may be fully charged in the first hour and fully discharged in the fifth hour. This will not affect the final cost. However, this phenomenon is unreasonable in scheduling. Therefore, it is appropriate to add some small costs to the charging and discharging of the ESS to resolve this. Where $COST_{batt}$ represents the cost of the battery ESS per unit of charge and discharge. In this study, it is set to 0.1 NTD/kWh

$$C_{batt}(t) = \left(P_{batt_{dis}}(t) + P_{batt_{ch}}(t)\right) \times COST_{batt} \tag{11}$$

Equation (12) is mainly to calculate the power of the battery ESS at time $t$. $SOC(t)$ represents the state of charge(SOC) of ESS at time $t$, $\delta_t$ is the time interval, $\eta_{Ind}$ and $\eta_{Inc}$ represent discharge and charge efficiency of the ESS, respectively, and $P_{batt\_capacity}$ represents the capacity of the ESS.

In inequality (13), $SOC_{min}$ and $SOC_{max}$ represent the minimum and maximum value of SOC. Table 1 shows the specifications of the different ESSs used for this study. As seen in the table, since different energy storages have different capacities and power characteristics, the two energy storages will not be able to charge or discharge full power when they are close to their full energy or almost no energy [27,28].

**Table 1.** ESS specification.

|  | Type | Specification | Original Function | $SOC_{min}$ | $SOC_{max}$ |
|---|---|---|---|---|---|
| ESS1 | lithium ion | 2 MW/1 MWh | frequency regulation | x | x |
| ESS2 | sodium-sulfur | 1.8 MW/10.8 MWh | energy arbitrage | 19% | 81% |
| ESS3 | not yet announced | 4 MW/24 MWh | energy arbitrage | 19% | 81% |

Originally, the $SOC_{min}$ and $SOC_{max}$ for both ESS2 and ESS3 are 10 and 90%, respectively. However, for this study, it would be set to 19 and 81%, respectively. These two values were adjusted because when an N-1 contingency occurs, the energy storage takes about half an hour (for the case of Kinmen Island) of continuous charging or discharging before a new generator is turned on. This charge or discharge decreases the SOC by 8.33%. Therefore, in order for the ESS2 and ESS3 to join the spinning reserve, their $SOC_{min}$ should be set to 19% and $SOC_{max}$ is set to 81% to make sure that the 8.33% of SOC for the N-1 contingency is always on standby and available at any time period.

In Equation (14), the final SOC must return to its initial value. $SOC_{ini}$ and $SOC_{end}$ represent the initial and end SOC, respectively.

$$SOC(t) = SOC(t-1) - \delta_t \left( \frac{P_{batt_{dis}}(t)}{\eta_{Ind} \times P_{batt_{capacity}}} + \frac{\eta_{Inc} \times P_{batt\_ch}(t)}{P_{batt\_capacity}} \right) \tag{12}$$

$$SOC_{min} \leq SOC(t) \leq SOC_{max} \tag{13}$$

$$SOC_{ini} = SOC_{end} \tag{14}$$

### 3.4. Power Balance Constraint

Shown in Equation (15) is the power balance constraint of the power system. Where $P_{PV}(t)$ represents the total power generation of renewable energy at time $t$. $P_{tolerance}(t)$ represents the allowable error value for the solution. $P_L(t)$ is the total load at time $t$.

$$P_{PV}(t) + P_{Inv}(t) + \sum_{n=1}^{N} P_n(t) + P_{tolerance}(t) = P_L(t) \tag{15}$$

### 3.5. Spinning Reserve Constraint

In inequality (16), the ESS available power is added to the spinning reserve to improve the reliability of the power grid. $P_{n\_max}(t)$ in (12) represents the maximum power generation of the $n$th generator while $P_{spin\_reserve}$ is the required standby capacity of the overall system. The current maximum generating capacity of the system at time $t$ is used as the reserve capacity limit, representing the left side of the Equation (16).

$$\sum_{n=1}^{N} U_n(t)(P_{n\_max}(t) - P_n(t)) + P_{batt\_dis\_max}(t) - P_{batt_{dis}}(t) + P_{batt_{ch}}(t) \geq P_{spin\_reserve} \tag{16}$$

### 3.6. Ramp Rate Constraint

In inequality (17), $R_{rate\_n}(t)$ represents the ramp rate of the nth diesel generator per second at the $t$th time. $R_{rate\_min}$ represents the minimum required ramp-up and ramp-down per second of the grid.

$$\sum_{n=1}^{N} U_n(t)R_{rate\_n}(t) \geq R_{rate\_min} \tag{17}$$

### 3.7. PV Curtailment Constraint

Curtailment of generated power is required when the penetration rate is too high. The scheduled PV generation needs to be less than the predicted generation, as shown in (18).

$$0 \leq P_{PV}(t) \leq P_{PV\_predict}(t) \tag{18}$$

### 3.8. Security Constraints

ESS charging power is used to increase the minimum frequency when the grid is vulnerable. The inequality is shown in Equation (19). $P_{batt\_ch\_ESS2}(t)$ and $P_{batt\_ch\_ESS3}(t)$ represent the respective charge amounts of the two ESSs specifically used for energy arbitrage. The $P_{constraint\_batt\_ch}(t)$ represents the minimum total charge required to support the N-1 contingency for these two ESSs at time $t$.

$$|P_{batt\_ch\_ESS2}(t)| + |P_{batt\_ch\_ESS3}(t)| \geq |P_{constraint\_batt\_ch}(t)| \tag{19}$$

## 4. Description and Introduction of Simulation Environment

### 4.1. System Model and Settings

This study takes the Kinmen grid as the system under study. Kinmen Island is a small island west of Taichung City (R.O.C.), Taiwan, very close to mainland China. The winter load is about 21.9 to 42.95 MW while the summer load is about 43.26 to 73.81 MW.

Kinmen Island uses diesel to generate electricity making its cost usually higher because of the cost of transporting fuel. On the other hand, solar energy is a cheaper replacement for fossil fuel.

Currently Kinmen Island has two power plants and a 12.3 MW PV plant. Power plant 1 has 10 heavy oil diesel generators. Power plant 2 has 6 light oil generators and 2 ESSs. ESS1 is 2 MW/1 MWh lithium-ion batteries used for frequency regulation, while ESS2 is 1.8 MW/10.8 MWh sodium-sulfur batteries used for energy arbitrage. It is expected that an additional 4 MW/24 MWh ESS will be built in 2023 for energy arbitrage.

This study considers the future winter conditions of Kinmen Island. Only heavy oil diesel generators will be used because of the low operational cost of heavy oil diesel generator. There is a 27 MW PV plant with the three ESSs mentioned in the previous paragraph. The system has 4–22.8 kV busbars, 4–11.4 kV busbars, 4 main transformer loads, and 2 substations. A simplified schematic diagram of connections between all facilities is shown in Figure 3. The trip settings for the underfrequency relays has four levels, 57.3, 57.0, 56.5, and 56.0 Hz. After triggering the underfrequency relay, it takes about 5–6 cycles to open the circuit breaker [5,29]. In this study, $F_{nadir}$ is set to 57.3 Hz, in order not to trigger UFLS.

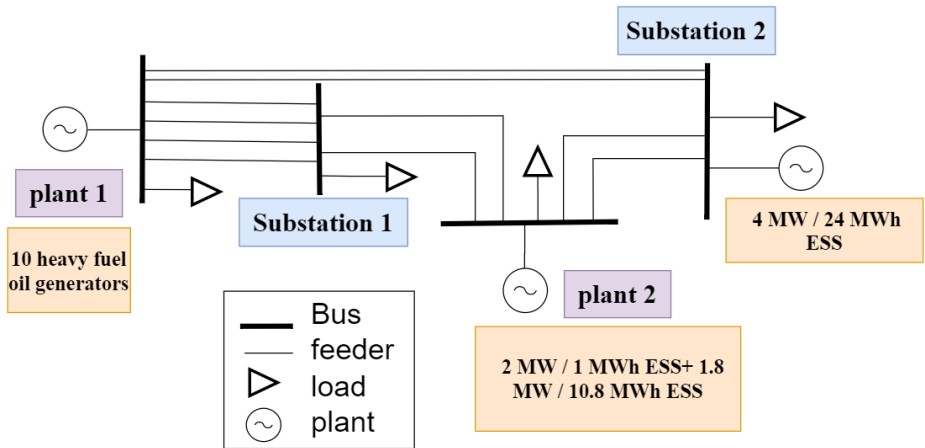

**Figure 3.** System diagram.

The ED is solved using mixed integer linear programming (MILP) using the IBM CPLEX 12.10.0 solver. The computer used is for these simulations is an Intel Core (TM) i5-7500 CPU @ 3.4 GHz. 16G RAM. PSS®E version is 33.4.0. It is coded in a Python program for automated simulation.

### 4.2. Generator ED Model

Various parameters of diesel generators and upper and lower limits of power generation in ED is shown in Table 2.

**Table 2.** Parameters of diesel generators.

| Units | Capacity (MVA) | Minimum Power Generation (MW) | Maximum Power Generation (MW) |
|---|---|---|---|
| Plant1 #1–4 | 10.2 | 4 | 7.7 |
| Plant1 #5–8 | 9.7 | 4.1 | 7.8 |
| Plant1 #9–10 | 13.8 | 5.5 | 10.5 |
| Plant2 #1–6 | 4.36 | 1.5 | 3.488 |

The fuel cost coefficients $a_n$, $b_n$, and $c_n$ of the hypothetical heavy oil diesel generator are shown in Table 3. The generator fuel cost after piecewise linearity is shown in Table 4.

**Table 3.** Diesel generator cost factor and startup cost.

| Units | $a_n$ | $b_n$ | $c_n$ | Start-Up Cost |
|---|---|---|---|---|
| Plant1 #1–4 | 15 | 1.9161 | 0.0661 | 7 |
| Plant1 #5–8 | 13 | 1.8518 | 0.0657 | 7 |
| Plant1 #9–10 | 12 | 1.7966 | 0.0615 | 10 |

**Table 4.** Diesel Generator Linear Fuel Cost.

| Units | Segment 1 | Segment 2 | Segment 3 |
|---|---|---|---|
| Plant1 #1–4 | 2.526x + 13.617 | 2.690x + 12.763 | 2.853x + 11.709 |
| Plant1 #5–8 | 2.471x + 11.564 | 2.634x + 10.699 | 2.796x + 9.635 |
| Plant1 #9–10 | 2.576x + 9.576 | 2.781x + 8.107 | 2.986x + 6.296 |

According to the government website [30], it is assumed that the ramp-up and ramp-down rate per sec of each generator set is shown in Table 5. In this study, this value was set to 430 kW/s. Since the highest ramp-up and ramp-down value of a single generator to 420 kW/s, the value of 430 kW/sec is chosen here to ensure that at least two generator sets will be running at any point in time. Two generators are needed because if the system trips contingency, at least one generator can provide the reactive power required for the grid to maintain voltage stability.

**Table 5.** The ramping rate of the generator set.

| Units | Ramping Rate (kW/s) |
|---|---|
| Plant 1 #1–4 | 15 |
| Plant 1 #5–8 | 15 |
| Plant 1 #9–10 | 420 |

### 4.3. Selection of Diesel Generator Model in PSS®E

The generator model selected in PSS®E is as shown in Table 6. All ESS models use second generation ESS general model of Western Electricity Coordinating Council (WECC). It consists of REPC_A, REEC_A and REGC_A.

**Table 6.** Diesel Generator model in PSS®E.

| Units | Dynamic Models | Excitation System Model | Governor Model |
|---|---|---|---|
| Plant1 #1–4 | GENSAL | ESAC8B | DEGOV |
| Plant1 #5–8 | GENSAL | IEEEX1 | DEGOV1 |
| Plant1 #9–10 | GNSAE | AC7B | DEGOV1 |

### 4.4. ESS's Response in Trip Contingency

Different ESSs will respond differently when a N-1 contingency fault occurs. The following will show the grid frequency and the actual power output of the two ESSs when a N-1 contingency fault occurs.

#### 4.4.1. ESS1(Frequency Regulation)

ESS1 is used for frequency regulation. When the frequency exceeds the deadband (59.85–60.12 Hz), the ESS will start to discharge power. The rising time from 0 to full output is 167 ms. The response is shown in Figure 4.

Using the calibrated generator parameters set in Section 4.3, as also use in the paper [29], the ESS1 response can be replicated using the PSS®E, as shown in Figure 5. Figure 5 shows the data measured during an N-1 contingency when the load is 36.3 MW on 13 December 2019. At this time, the ESS1 and ESS2 has not been completed. Figure 6 shows the frequency measurement and simulation results for this N-1 contingency. A good match exists of $F_{nadir}$ between calibrated simulation and measurement.

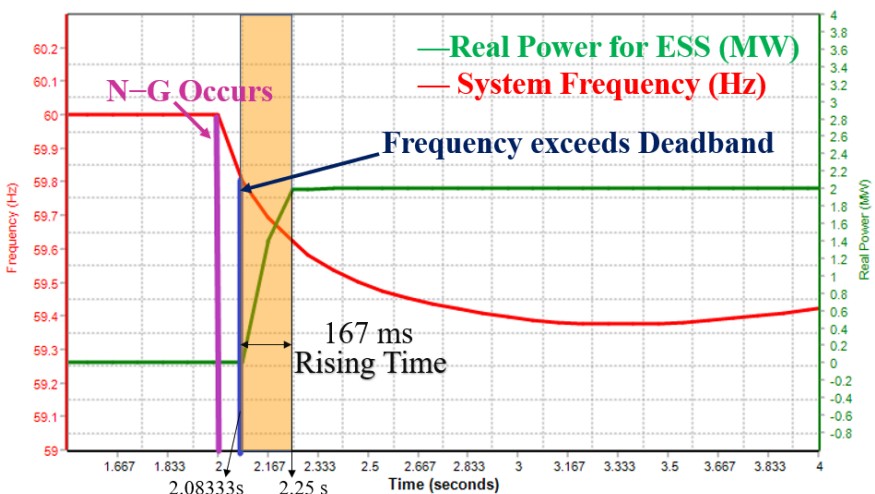

**Figure 4.** ESS1 real power output and grid frequency when N-1 contingency occurs.

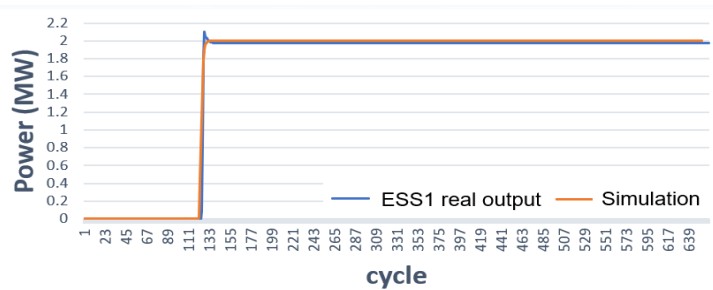

**Figure 5.** Output response of ESS1 to an N-1 contingency from PSSE and as measured from phasor measurement Unit (PMU).

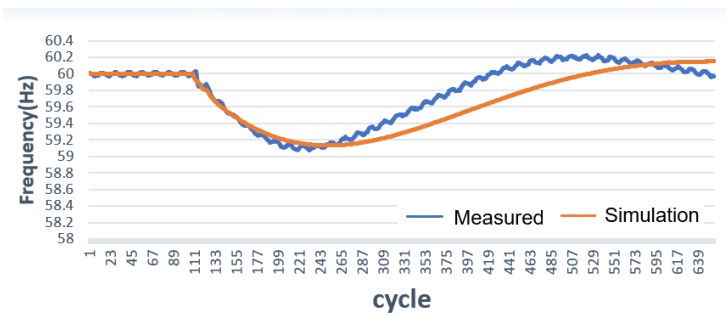

**Figure 6.** Frequency response simulation (PSSE) of ESS2 and ESS3 to a N-1 contingency and as measured from PMU.

### 4.4.2. ESS 2 and ESS 3 (Energy Arbitrage)

The responses of ESS2 and ESS3 during charging and discharging are as follows:

When the ESS is charging and a fault occurs, the ESS will quickly stop charging using low frequency relay tripping. As shown in Figure 7.

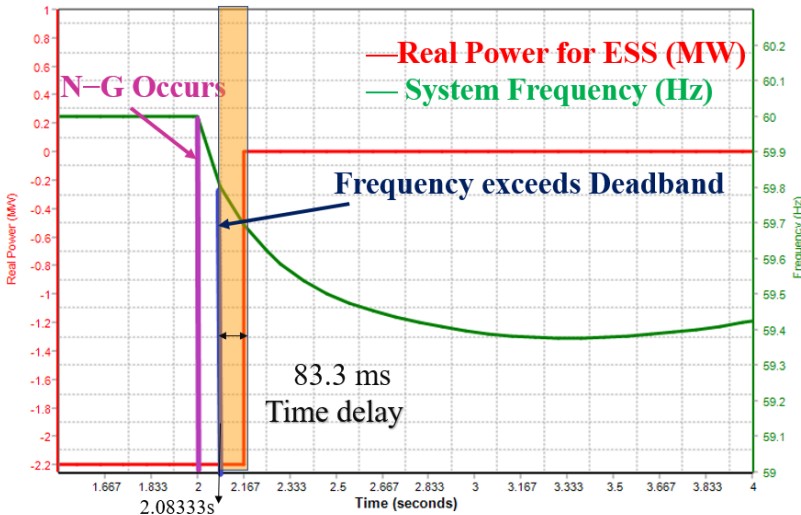

**Figure 7.** ESS 2 and ESS 3 grid frequency and real power output when N-1 contingency occurs while charging.

When the ESS is discharging and a fault occurs, the ESS will keep discharging. As shown in Figure 8.There would be a time delay of 83.33 ms from exceeding the deadband (59.85–60.12 Hz) to cutting off the ESS.

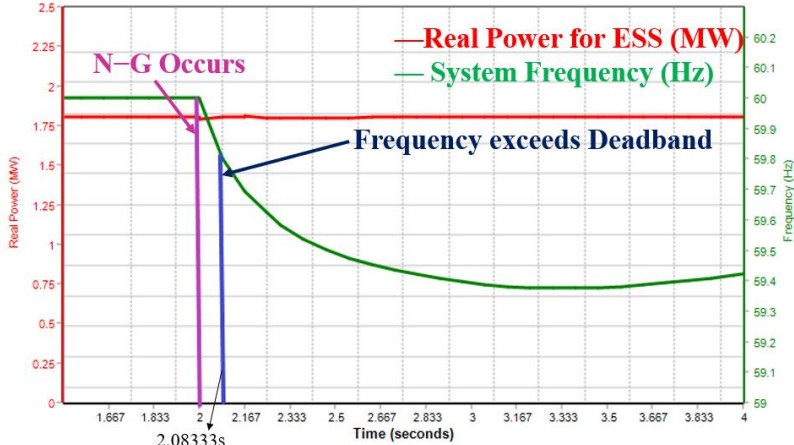

**Figure 8.** Grid frequency and real power output when contingency occurs during ESS2 and ESS3 discharge.

When ESS2 and ESS3 are discharging and the contingency occurs. As long as the voltage does not exceed the allowable range of high voltage ride through (HVRT) and low voltage ride through (LVRT), the ESS will continue to discharge and not cut off from the grid.

The transient performance of ESS2 and ESS3 depends on the circuit breaker, it is assumed that the local circuit breaker can cut off the electricity after about 5 cycles [27]. It can be seen from the picture that the simulated energy storage transient output and the lowest frequency of the contingency have a good match with the actual measurement.

## 5. Simulation Results

### 5.1. Case 1: Multi-Function ESS (Proposed Method)

Figure 9 show the results of the first ED or the initial ED. The N-1 contingency minimum frequency per hour is shown in Figure 10 with a blue line. Two generators are operating on at the 9th hour, with values of 6.4 and 7.1 MW. In the 15th hour, two

generators are operating with output of 6.6 and 7.1 MW. The N-1 contingency minimum frequency for the 9th and 15th hours are 56.079 and 56.074 Hz, respectively.

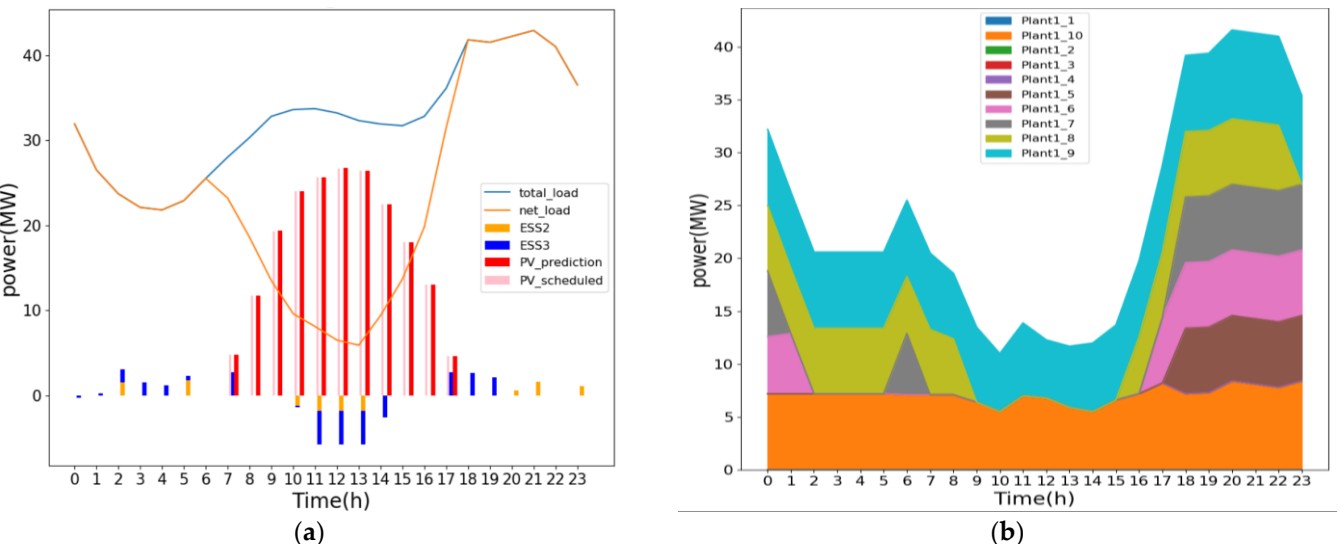

**Figure 9.** Case 1: result of first ED of (**a**) PV generation and the charge and discharge of ESS and (**b**) the power output of the diesel generators.

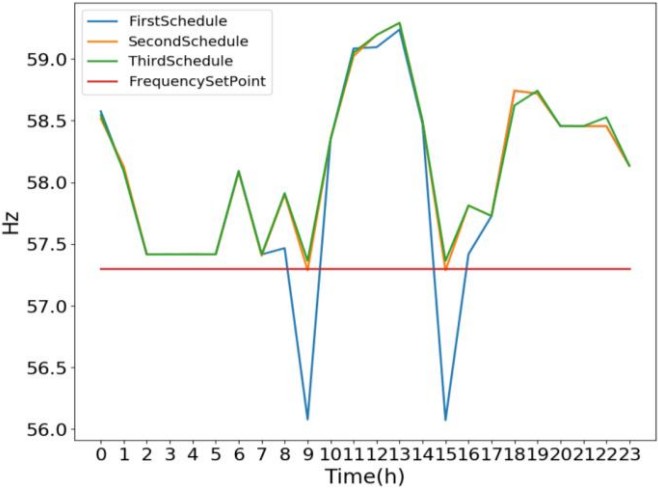

**Figure 10.** Minimum frequency of N-1 contingency per hour for three schedules of case 1.

The 9th hour frequency simulation results with generator active power outputs are shown in Appendix A. The ESS2 and ESS3 were not charged in both hours. The total cost of this third ED is 2422 kNTD. Using the process flow as discussed in Figure 1, the ESS requires 1.7 MW to charge at 9th hour and 1.7 MW to charge at 15th hour to get the frequency above the set value, equivalent to the 8.33% of SOC. Adding these two limits for the said hours, the second ED is rescheduled.

The results of the second ED are shown in the Figure 11. Case 1: result of second ED of (a) PV generation and the charge and discharge of ESS and (b) the power output of the diesel generators.

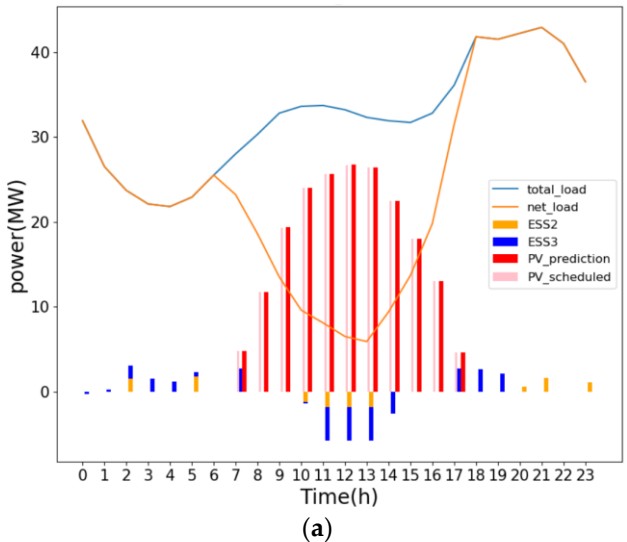
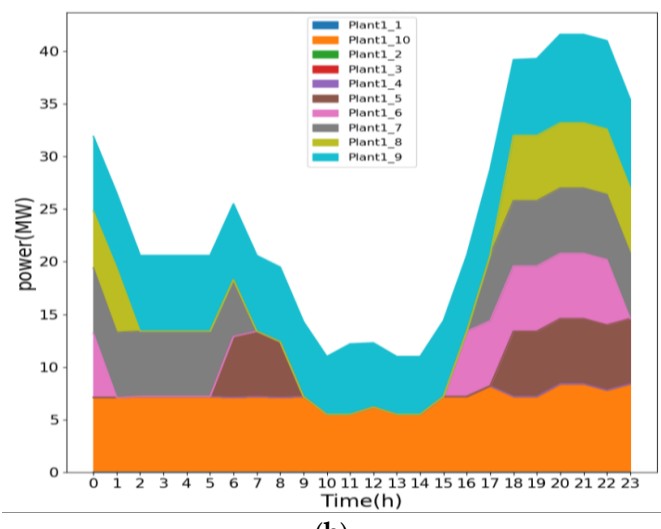

(**a**)　　　　　　　　　　　　　　　　　　　(**b**)

**Figure 11.** Case 1: result of second ED of (**a**) PV generation and the charge and discharge of ESS and (**b**) the power output of the diesel generators.

The minimum frequency of the N-1 contingency per hour is shown in Figure 10 in the orange line. Notice that the minimum frequency touches the set value of the frequency. The minimum frequency of N-1 contingency for the 9th hour and the 15th hour are 57.287 and 57.287 Hz, respectively. Using the proposed procedure to determine the required ESS charging for the 9th and 15th hours, the charging power required are 1.8 MW for the 9th hour and 1.8 MW for the 15th. Again, it is required to add this in the charging limits for the next ED. The total cost of this third ED is 2422 kNTD.

The result of the third ED is shown in Figure 12. The minimum frequency of N-1 contingency per hour is shown in Figure 10 in the green line. The 9th hour frequency simulation results with generator active power outputs are shown in the Appendix A. The minimum frequency of each hour is higher than the set value. Therefore, this is the final ED to support the N-1 contingency. The total cost of this third ED is 2422 kNTD.

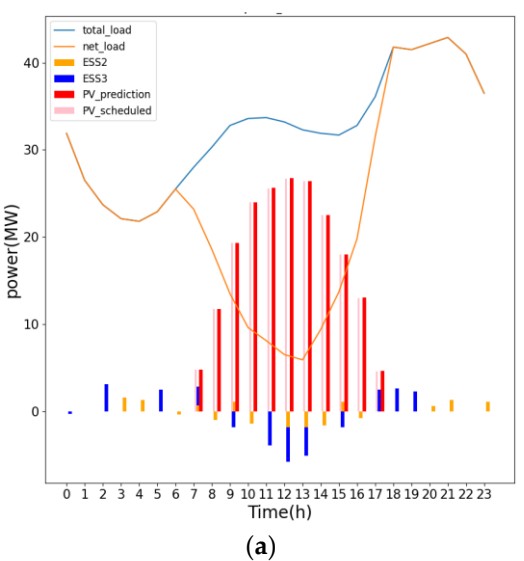
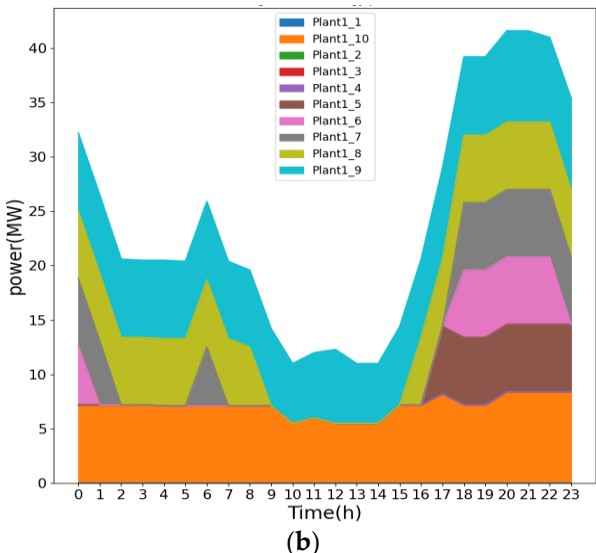

(**a**)　　　　　　　　　　　　　　　　　　　(**b**)

**Figure 12.** Case 1: result of third ED of (**a**) PV generation and the charge and discharge of ESS and (**b**) the power output of the diesel generators.

The Figure 13 shows the changes in ED of the ESS2 and ESS3. In the second and third results, the two energy storages are in the state of charge and discharge at the 9th and 15th hours, so that $F_{nadir}$ can be increased, and keep close to the original total output of that hour.

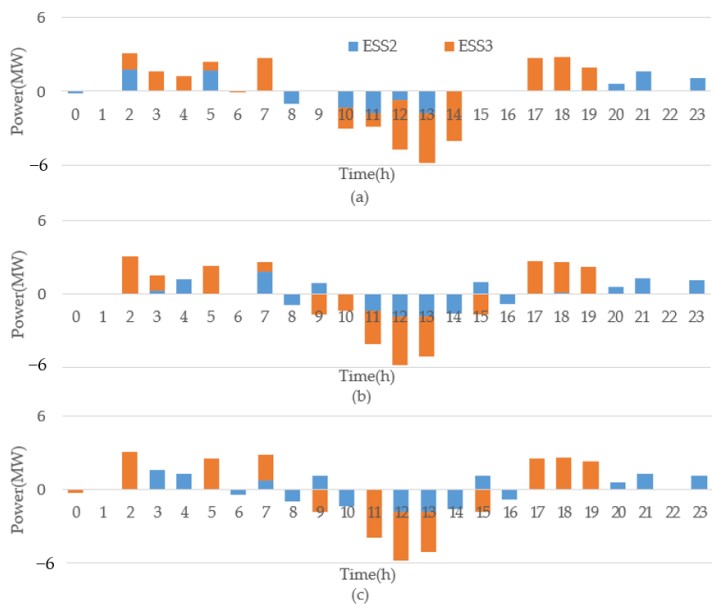

**Figure 13.** ED results of the ESSs (**a**)Result of 1st ED (**b**) Result of 2nd ED (**c**) Result of 3rd ED.

### 5.2. Case 2: ESS Functioning as a Frequency Support

Removing the two ESS, that function as energy arbitrage, verifies energy arbitrage function of our proposed method. The Figure 14 shows the ED result when the ESS2 and ESS3 are removed. According to the system conditions, two generators must be turned on to maintain the system stability.

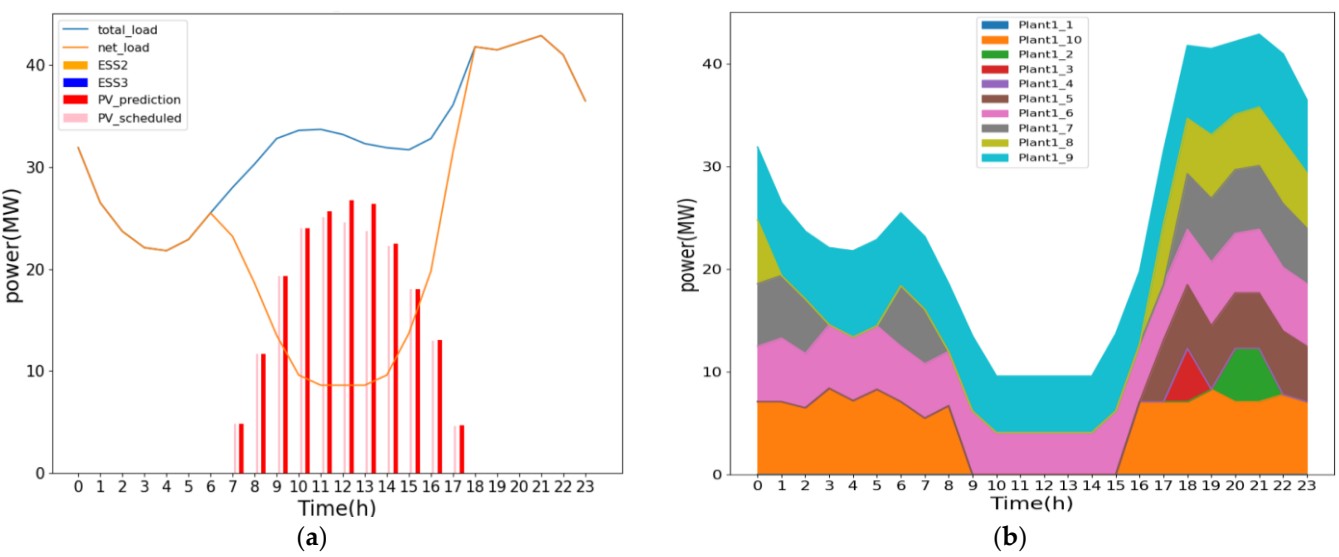

**Figure 14.** Case 2: Result of the scenario without ESS for (**a**) the PV generation and (**b**) the power output of the diesel generators.

Since there are no ESS performing energy arbitrage, low net load causes the PV power generation to be curtailed, as shown in the Figure 14a, as the pink bar is lower than the red

bar. Furthermore, without energy arbitrage, the total operating cost raises to 2555 kNTD from 2422 kNTD.

## 6. Discussion

It can also be found from the simulation results that when N-1 contingency occurs, the most dangerous instance is not necessarily when the penetration of renewable energy is the highest, but tends to occur in the midway from zero penetration to highest penetration.

The reason may be that when the penetration of renewable energy at its peak, the ESS will be charged more. If a contingency occurs, it can quickly trip and reduce a lot of energy use. The lowest point of the frequency is related to the amount of power generation that is tripped. When the penetration rate is the highest, the generators are almost always lightly loaded, so the reduced power generation of the contingency is relatively small. Therefore, even if the system inertia at that time is small, the lowest frequency may not be the lowest during this period.

Furthermore, in theory the outcome of stopping the charging of EVs would be similar to cutting off the ESS from charging in this study, probably also increasing the minimum frequency of N-1 1 contingency. The results of this study can be used to set the time price of electric vehicles to charge, so that more EVs can be charged when the power grid is weak.

In the picture., two ESSs are used to charge and discharge, respectively, to improve the safety of the N-1 contingency. In the real world, energy loss will occur due to the round-trip efficiency of the ESS. This situation can be understood as exchanging energy for N-1 contingency resiliency. May be especially suitable for pumped-storage power plants when rainwater is abundant.

## 7. Conclusions

This paper proposes a method by which energy arbitrage energy storage can help the N-1 contingency. The frequency regulation ESS and the energy arbitrage ESS are considered in the simulation. PSS®E is used to verify that the energy arbitrage ESS disconnected from charging can increase the minimum frequency when contingency occurs. In this way, the ESS can provide spinning reserve, energy arbitrage, and help N-1 contingency at the same time. This method is also not only suitable for lithium-ion batteries, but for all battery types. It can also be applied to various ESSs, such as flow batteries, pumped storage power, etc. The simulation results show that the proposed method can effectively improve the minimum contingency frequency higher than the set value.

**Author Contributions:** Conceptualization, T.-C.T.; Methodology, J.-Z.J.; Software, J.-Z.J. and P.-Y.C.; Validation, P.-Y.C.; Visualization, T.-C.T.; Supervision, C.-C.K.; Project administration, C.-C.K.; Funding acquisition, C.-C.K. All authors have read and agreed to the published version of the manuscript.

**Funding:** The support of this research by the Ministry of Science and Technology of the Republic of China under Grant No. MOST 111-2622-8-011-006-TE1 & MOST 111-3116-F-006-006—are gratefully acknowledged.

**Institutional Review Board Statement:** Not applicable.

**Informed Consent Statement:** Not applicable.

**Data Availability Statement:** All data are provided in this manuscript.

**Conflicts of Interest:** The authors declare no conflict of interest. The funders had no role in the design of the study; in the collection, analyses, or interpretation of data; in the writing of the manuscript, or in the decision to publish the results.

## Appendix A

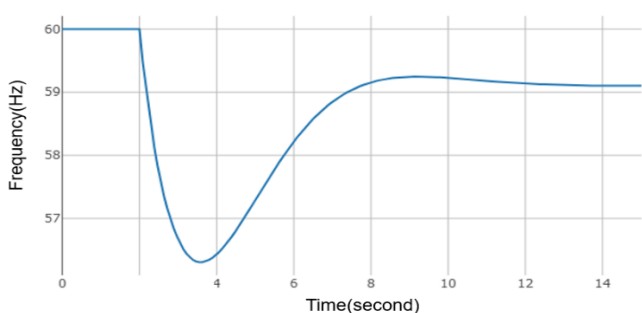

**Figure A1.** N-1 contingency frequency at 9th hour of first ED.

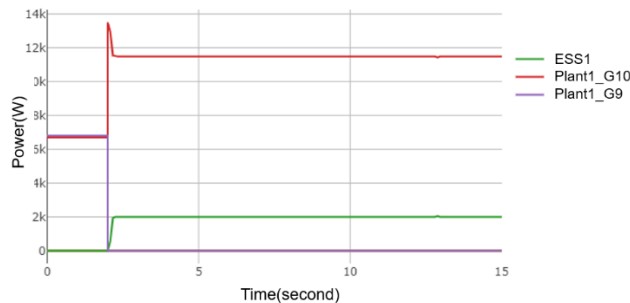

**Figure A2.** Generators active power at the 9th hour of the first ED.

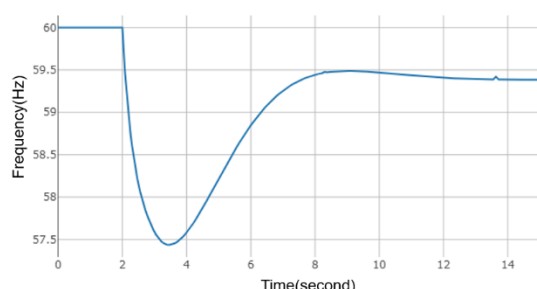

**Figure A3.** N-1 contingency frequency at 9th hour of 3rd ED.

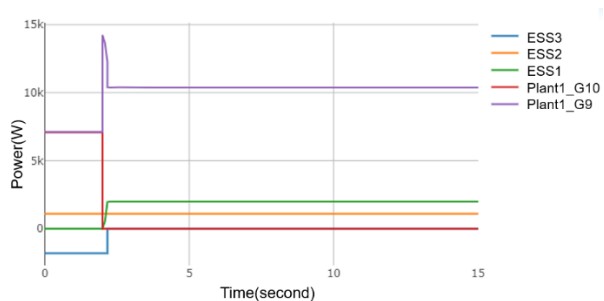

**Figure A4.** Generator power at the 9th hour of the 3rd ED.

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
