# Peer review of "Research on Dynamic Reserve and Energy Arbitrage of Energy Storage System"

_applsci, doi:10.3390/app122311953_

Round 1

Reviewer 1 Report

I recommend to the authors implement some experimental analysis results to the paper in order to improve the paper quality. 

1. Conclusion should be improved in terms of the simulation findings.

2. Abstract is very short. It should also covers the main idea of the paper clearly.

3. There are many grammar mistakes in the paper. Language should be improved.

4. There is no experimental results in the paper. It only covers simulation results. I recommend that At least, authors should compare the obtained simulation results with experimental results in any paper in literature by providing reference paper. 

5. Major.

Reviewer 2 Report

The manuscript aims at proposing a method for the ESS to simultaneously provide energy arbitrage, reserve capacity, and assist N-1 contingency.

The manuscript is difficult to read not due to the english but due to its organization. The main contribution of the manuscript would be the iterative optimization until the minimum frequency in an N-1 contingency case is not lower than a defined setpoint. However it is not detailed how this minimum frequency is estimated.

Why is F (the objective function) in equation 2? Or is it that in line 81 instead of Cn, you meant FCn? Where does equation 2 comes from?

In Fig. 1 what do you mean by "add 0.1 MW of charging capacity"? It seems that you change the power capacity of the battery, which would mean changing the battery itself. Do you perhaps refer to the 'charging power setpoint' instead of capacity?

The explanation about the line segments in line 85-91 is not clear at all. You use i for the i-th generator, but also for the number of the line segment. Is not clear what the line segment is. Then you relate alfa and beta in equation 6 to these line segments. You could include a figure that puts into relation all these factors (the line segment, the quadratic function, alfa, beta), and clarify the employ of binary variables Bn_i. Indeed, how the cost expressed in (2) influences the fuel cost in (6)?

In line 126 authors state "Continuous charging and discharging for half an hour will increase or decrease the power by about 8.33 %." First of all, authors probably refer to SOC instead of power. How come do they consider half hour if it seems that the optimization is run in periods of 1 hour? Where does this 8.33% comes from? You could include some data of the power and capacity of the battery (particularly the maximum discharging and charging power) before introducing this calculation.

In line 127 authors talk about ESS2 and ESS3, but they are not introduced until the next section, which is confusing.

Regarding equation (16), can authors comment on what happens when the battery is very low or even completely discharged? As this factor is not considered in (16) since Pbatt_dis and Pbatt_ch consider only the instant discharging or charging power of the battery, but not its state of charge.

While the results might be interesting, they are not clearly presented, as it is difficult for the reader to understand the differences between one case and the other. In line 262 authors state "The total cost of this third ED is 2,422kNTD" but the reader has no way of comparing this to results without the proposed strategy. Please include the costs of the first and second EDs also. Also, you should make a figure with the profiles of the ESSs in all the cases, so we can appreciate how much they change.

A transitory simulation in the critical hours (e.g, at 9 or 15) of a contingency would be required to verify that effectively the frequency does not drop below the setpoint level.

A main question: How is the minimum frequency F_nadir calculated? If it is in references [19,20] as stated in line 50, please include some principles here, as it is a central part of the strategy.

The manuscript contains redaction issues, like punctuation. Some examples of sentences that are not clear (not the only cases):

line 28: "...but with peak shaving ESS is not considered."

line 39: "the ESS will reduce the minimize the cost"

line 58: "The scheduled ED will be used to determine the minimum frequency (F_nadir) of the maximum unit trip contingency will be calculated at each hour in the PSS®E."

Reviewer 3 Report

Literature review must be improved. The list of journals papers should be expanded, especially with more recent articles. There are few papers published after 2020. I recommend improving the graphic presentation of the figures.

Round 2

Reviewer 2 Report

Thank you for considering my concerns.